# Effectiveness of Vaccination against SARS-CoV-2 Infection in the Pre-Delta Era: A Systematic Review and Meta-Analysis

**DOI:** 10.3390/vaccines10020157

**Published:** 2022-01-21

**Authors:** Angela Meggiolaro, Monica Sane Schepisi, Georgios F. Nikolaidis, Daniele Mipatrini, Andrea Siddu, Giovanni Rezza

**Affiliations:** 1Italian Ministry of Health, General Directorate for Health Prevention, Viale Ribotta 5, 00144 Rome, Italy; m.saneschepisi@sanita.it (M.S.S.); d.mipatrini@sanita.it (D.M.); a.siddu@sanita.it (A.S.); g.rezza@sanita.it (G.R.); 2IQVIA, 210 Pentonville Rd, London N1 9JB, UK; georgios.nikolaidis@iqvia.com

**Keywords:** coronavirus disease 2019, SARS-CoV-2 infection, vaccine effectiveness

## Abstract

(1) Background: The objective of this study was to assess the effectiveness of SARS-CoV-2 vaccines in terms of prevention of disease and transmission in the pre-Delta era. The evaluation was narrowed to two mRNA vaccines and two modified adenovirus-vectored vaccines. (2) Methods: The overall risk of any SARS-CoV-2 infection confirmed by positive real-time Polymerase Chain Reaction (PCR) test was estimated in partially and fully vaccinated individuals. The evidence synthesis was pursued through a random-effects meta-analysis. The effect size was expressed as relative risk (RR) and RRR (RR reduction) of SARS-CoV-2 infection following vaccination. Heterogeneity was investigated through a between-study heterogeneity analysis and a subgroup meta-analysis. (3) Results: The systematic review identified 27 studies eligible for the quantitative synthesis. Partially vaccinated individuals presented a RRR = 73% (95%CI = 59–83%) for positive SARS-CoV-2 PCR (RR = 0.27) and a RRR=79% (95%CI = 30–93%) for symptomatic SARS-CoV-2 PCR (RR = 0.21). Fully vaccinated individuals showed a RRR = 94% (95%CI = 88–98%) for SARS-CoV-2 positive PCR (RR = 0.06) compared to unvaccinated individuals. The full BNT162b2 vaccination protocol achieved a RRR = 84–94% against any SARS-CoV-2-positive PCR and a RRR = 68–84% against symptomatic positive PCR. (4) Conclusions: The meta-analysis results suggest that full vaccination might block transmission. In particular, the risk of SARS-CoV-2 infection appeared higher for non-B.1.1.7 variants and individuals aged ≥69 years. Considering the high level of heterogeneity, these findings must be taken with caution. Further research on SARS-CoV-2 vaccine effectiveness against emerging SARS-CoV-2 variants is encouraged.

## 1. Introduction

Since December 2020, infections with SARS-CoV-2 and the associated disease, COVID-19 (coronavirus disease 2019), have spread worldwide. On 11 March 2020, the WHO declared the COVID-19 outbreak a pandemic. The coronavirus disease 2019 (COVID-19) has represented a serious threat to public health, with nearly 4,038,342 deaths and 186,821,815 confirmed cases reported globally. As of 5 July 2021, 3,114,766,865 vaccine doses had been administered [1].

Coronaviruses (CoVs), including SARS-CoV, MERS-CoV, and SARS-CoV-2, are positive-sense, single-stranded RNA viruses with four structural proteins (S protein, envelope (E) protein, membrane (M) protein, and nucleocapsid (N) protein) [2]. The spike protein of SARS-CoV-2 binds to ACE2 receptors on target cells and acts as an immunodominant antigen, eliciting both antibody and T-cell responses [3].

To date, the most important COVID-19 candidate vaccines have been developed by using mRNA technology, adenoviral vectors, inactivated virus, and adjuvants [4]. During the COVID-19 pandemic, the European Medicines Agency (EMA) has applied a conditional marketing authorization for the fast-track approval of safe and effective COVID-19 vaccines in the EU. Based on RCT results, the EMA recommended three COVID-19 vaccines to prevent COVID-19: Comirnaty (BNT162b2), Vaxzevria (ChAdOx1/AZD1222), and Moderna (mRNA-1273). In March 2021, the EMA further expanded the arsenal of COVID-19 vaccines available to European member states with the COVID-19 vaccine, Janssen (Ad26.COV2.S) [5].

However, while efficacy estimates, in terms of the degree to which a vaccine prevents disease and/or transmission, are measured under ideal circumstances, such as Controlled Randomized Trials (RCTs), effectiveness refers to an estimate of the vaccine’s performance in the real world. In general, effectiveness is measured through observational studies, in which participants are not randomly assigned to an intervention versus a control group [6]. The effort to estimate vaccine effectiveness through meta-analysis meets decision makers’ requirements regarding different issues. First, the synthesis of the best available evidence can support public health prevention strategies and encourage vaccine uptake. Second, meta-analyses provide indirect information regarding the evolution of SARS-CoV-2’s pathogenicity and virulence in response to mass vaccination. In particular, they produce indirect evidence on the potential consequences of SARS-CoV-2 vaccine escape in terms of morbidity and mortality. Finally, yet importantly, monitoring the adaptation of SARS-CoV-2 to growing mass immunity can address further therapeutical or prevention strategies against SARS-CoV-2 variants of concern [7,8].

The main objective of this study was to estimate the effectiveness of conditional market-licensed COVID-19 vaccines in terms of the prevention of disease and transmission. The vaccine effectiveness (VE) was measured as relative risk (RR) and relative risk reduction (RRR) of any SARS-CoV-2 infection, confirmed by polymerase chain reaction (PCR) test, on partially and fully vaccinated individuals compared to unvaccinated individuals. The meta-analysis estimated the overall RR of asymptomatic and symptomatic SARS-CoV-2 infections after partial and full vaccination protocol. The evaluation was narrowed to two mRNA vaccines (Comirnaty and Moderna), a modified adenovirus vaccine (COVID-19 Vaccine AstraZeneca or Vaxzevria), and a recombinant human adenovirus vector (Ad26.COV2.S or Janssen or J&J).

## 2. Methods

The systematic review started on March 2021 and concluded on 15 May 2021. The main purpose of this research was to assess the risk of COVID-19 occurrence (any positive RT-PCR test) among vaccinated subjects. Moreover, we aimed at quantifying the risk of developing symptomatic COVID-19 after vaccination. The evaluation was narrowed to the BNT162b2, mRNA-1273, ChAdOx1/AZD1222, and Ad26.COV2.S vaccines. The study followed the PRISMA 2020 statement [9]. The systematic review was prospectively registered on PROSPERO (n. CRD42021240143).

### 2.1. Search and Selection

The search strategy is presented in detail in Appendix A. We used seven different web engines, including early-stage research platforms: PubMed, Cochrane, clinicaltrial.gov, COVID-NMA, medRxiv, SSRN, and Authorea. No restriction on language, setting or publication date was imposed.

The papers were selected initially according to their titles and abstracts. The full texts that were suitable for the quantitative synthesis were collected in an Excel database for the data extraction. The exclusion criteria included lack of suitable data and study design. We narrowed the quantitative synthesis to non-experimental studies only. 

### 2.2. Data Extraction

The data extraction was performed by two authors, independently. The results of the respective phase III RCTs were used as references for data extraction [6,7,10,11]. The information derived from each full text was classified into four categories: (1) outcome, (2) study characteristics (design, publication status, year of publication), (3) participants characteristics (mean age, severity of COVID-19 symptoms, dose of SARS-CoV-2 vaccine, SARS-CoV-2 lineage) and (4) risk of bias. Disagreements were resolved by discussion or, if necessary, consultation with a third party.

### 2.3. Outcome

The primary endpoint of this research was to measure the overall relative risk (RR) of any SARS-CoV-2 infection 14 days after the first dose (partially vaccinated), seven days after the second dose (fully vaccinated), and 14 days after at least one dose uptake.

As a secondary endpoint, we aimed to measure the risk of symptomatic infection following SARS-CoV-2 vaccination, and, if possible, to quantify the risk of hospitalization and death after partial and full vaccination protocol.

To this end, any COVID-19 infection, either symptomatic or asymptomatic, confirmed through PCR was considered as a COVID-19 case. Studies that did not feature the routine performance a PCR test after vaccination were excluded. Documented SARS-CoV-2 infections (any positive rtPCR) at the baseline were excluded from the effectiveness analysis.

The risk of infection between vaccinated and unvaccinated groups was calculated as relative risk (RR). In order to quantify how much SARS-CoV-2 vaccination reduced the risk of infection relative to the control group, we computed the relative risk reduction (RRR) according to the formula 100*(1- RR).

### 2.4. SARS-CoV-2 Infection Episodes

The primary analysis included the RR of any new positive infection episode among vaccinated and unvaccinated groups confirmed through PCR test. The impact of SARS-CoV-2 vaccination on infection severity was investigated in secondary analysis as the RR of symptomatic positive PCR and, possibly, the RR of hospitalization and death following partial and full vaccination protocols. Both self-reported symptoms and symptoms ascertained through clinical visit were included.

Finally, COVID-19 episodes were classified as compatible with the B.1.1.7 variant and incompatible (i.e., B.135, non-B.1.1.7, unspecified).

### 2.5. Experimental Group

All adults eligible to undergo SARS-CoV-2 mass vaccination were included in the experimental group. For the definition of cases in the vaccinated groups, we referred to Polack et al., Phase III RCT [6]. The same rules were applied to ChAdOx1/AZD1222 and mRNA-1273. Concerning ChAdOx1/AZD1222, only a small number of studies on VE after the first dose administration were available, as the vaccine was approved later for the emergency rollout [10]. The single-dose administration of Ad26.COV2.S was considered as full vaccination protocol, although the induction time lasts two weeks (Figure 1) [11].

The 14 days following the administration of the first dose were designated as induction time because they were considered as the minimum time spell needed to develop COVID-19 immunity. Therefore, regardless of vaccine technology, SARS-CoV-2 cases occurring within two weeks of the first dose were not considered as cases within the vaccinated population [6,12]. In order to assess the effectiveness of the full vaccination protocol, the upper follow-up limit was set at seven days after the second dose. Hence, only SARS-CoV-2 cases occurring at least one week after the second dose were attributed to a lack of vaccine protection [6]. Moreover, cases occurring within these seven days were still attributed to the first dose’s effectiveness (Figure 1).

The effectiveness of the vaccines against positive SARS-COV-2 with at least one dose was tested 14 days after the administration of the first dose.

### 2.6. Control Group

In the general population, the mean incubation period for COVID-19 symptom onset was estimated as 5.8 days [13]. So far, vaccination has been prioritized towards older and at-risk individuals; this aspect might have biased the comparison with the unvaccinated cohort, which generally was younger and healthier. Overall, cohort studies drew the control group from the same population as the vaccinated group, while in a few cases the cohort was the same community observed before and after the vaccination. For reasons mostly related to a dearth of detailed information on unvaccinated individuals, the beginning of the follow-up for the control group was established at T0 (Figure 1).

### 2.7. Risk of Bias

In order to assess the risk of bias, we employed the Newcastle–Ottawa Scale (NOS) in cohort and case-control studies, and the NOS adapted for cross-sectional studies [14].

### 2.8. Statistical Analysis

We performed a frequentist meta-analysis using the inverse variance (IV) method [15]. In addition, the Mantel–Haenszel (MH) method was applied because we expected to encounter sparse data, especially in estimating the RR of symptomatic cases after vaccination. The analysis was executed on R (version 4.0.5).

The random effect (RE) meta-analysis was considered more appropriate to estimate the overall effect size. However, both fixed effects (FE) and RE outputs were reported. In order to test for the overall heterogeneity, both the Cochran’s Q (chi-squared statistic) and I-squared values (I^2^) were calculated. The Der Simonian and Laird method estimated the between-study variance τ^2^ [16].

As between-study heterogeneity can be caused by studies with extreme effect sizes, low-quality or small sample sizes, the following analyses were performed: outliers test, influence analyses [17], Baujat Plot analysis [18], leave-one-out analysis, and Graphic Display of Heterogeneity (GOSH) with Knapp–Hartung adjustment [19] (Appendix A). 

The subgroup meta-analysis, which enclosed vaccine type, quality, age of vaccinated population, and SARS-CoV-2 lineage detected through PCR tests, was performed to identify further heterogeneity sources [19]. As we expected a limited number of studies in some subgroups (*n* < 10), we opted for a FE model in the subgroup meta-analysis. The subgroup levels were considered fixed and exhaustive. Nevertheless, in order to capture discrepancies that might yield inconclusive results, more conservative methods, such as RE and mixed model, were added to the output (reported in Appendix A).

In order to test whether publication biases influenced the study conclusions, we performed a funnel plot (Appendix A) [20].

## 3. Results

The web search provided 7760 unduplicated records. Overall, 31 studies were selected for data extraction and qualitative data synthesis [21,22,23,24,25,26,27,28,29,30,31,32,33,34,35,36,37,38,39,40,41,42,43,44,45,46,47,48,49,50,51] (Appendix A). There were 24 cohort studies, two case-control, one cross-sectional, one matched observational, and three test negative case-control. Ten studies were performed in the UK, eight in the US, and five in Israel. The RR of SARS-CoV-2 infection among health care workers (HCWs) was examined by thirteen studies; four studies analyzed long-term care facility (LCTF) residents and nine the general population ( Appendix A).

### 3.1. Meta-Analysis

In the following meta-analysis, an RE model with inverse variance (IV) method was used throughout. The evidence synthesis was based on RR as the measure of effect. Additionally, the FE estimates and the Mantel–Haenszel outputs are reported in Table 1. The between-study variance τ^2^ was computed for each RE meta-analysis.

Overall, 27 out of 31 studies were included in the quantitative analyses. Four studies met the inclusion criteria for the systematic review but were not included in the meta-analyses [41,44,45,46].

Despite some small sample sizes and the small number of infection cases expected among vaccinated individuals, no study presented zero cells during the data extraction. The ChAdOx1/AZD1222 effectiveness was investigated in 4 out of 27 studies [22,23,42,43], while only one study estimated Ad26.COV2.S effectiveness [38].

The Newcastle–Ottawa Quality Assessment Scale showed a satisfactory quality score, with a median of six, a minimum of five, and a maximum of eight ( Appendix A).

#### 3.1.1. RR of SARS-CoV-2 Infection following Vaccination

The meta-analysis on first-dose VE against any positive PCR pooled 17 studies and 22 entries. Jones’ analysis spanned a follow-up period of two weeks (A) and, additionally, an extended period of six weeks (B) [21]. Lopez-Bernal et al. evaluated the effectiveness of one and two doses of the BNT162b2 vaccine on adults aged ≥80 years (A80 and B80) and ≥70 years (B70 and C70) in England. Additionally, the effectiveness of a single dose of ChAdOx1/AZD1222 (A70) and at least a single dose of BNT162b2 (C70) were tested on adults aged ≥70 years (A70) [22]. Shotri et al. estimated the protective effects of the first dose of BNT162b2 (A) and ChAdOx1/AZD1222 (B) against any SARS-CoV-2 infection [23]. One study from Denmark investigated BNT162b2 VE in two cohorts, namely long-term care facility residents (R) and HCWs (H) [25]. Abu Raddad et al. measured the effectiveness of BNT162b2 against B.1.1.7 (A) and B.1.351 (B), which became predominant within Qatar in early 2021, and any different (C) SARS-CoV-2 variants [25]. According to the RE meta-analysis results (IV), partially vaccinated individuals showed an RRR of 73% for any SARS-CoV-2 positive PCR compared to unvaccinated individuals (RR = 0.27). The FE model yielded lower effectiveness estimates after partial vaccination (RR = 0.41; RRR = 59%). Although both FE and RE (IV) showed a statistically significant result (*p* < 0.0001), the heterogeneity was considerable (τ^2^ = 1.08; H = 18.43 (17.52; 19.38; I^2^ = 99.7%); therefore, the true size of the effect remained uncertain (Table 1, Figure 2a).

In the evaluation of the RR for any positive SARS-CoV-2 PCR following the full vaccination protocol, only data on mRNA vaccines (BNT162b2 and mRNA-1273) were available. Sixteen studies and seventeen entries were included in this meta-analysis. One study evaluated Ad26.COV2.S effectiveness, while Pritchard indistinctly assessed the effectiveness of ChAdOx1/AZD1222 and BNT162b2 after the full vaccination protocol [26]. In Abu Raddad’s study, the data on PCR tests were available after full vaccination protocol with BNT162b2 (two weeks after second dose administration) [25]. Four studies did not distinguish between the BNT162b2 and mRNA-1273 vaccines [27,28,29,30].

Based on the RE meta-analysis results (IV), the RRR for any positive SARS-CoV-2 PCR following the full vaccination protocol was 94% (95%CI = 88–98%) with RR = 0.06, compared to unvaccinated individuals (Table 1, Figure 2b). FE yielded a RRR = 88%. Although both FE and RE showed a statistically significant protective effect of full COVID-19 vaccination (*p* < 0.0001), we should be cautious about drawing conclusions as to the true size effect because the heterogeneity was considerable (τ^2^ = 2.6; H = 10.07 (9.22; 11.00); I^2^ = 99.0%). The test for heterogeneity was significant at 1% (Q = 1621.96, *p* = 0). (Table 1, Figure 2b).

A longer follow-up period was available in the VE estimation after at least one dose. For this group, the mean length of follow-up was 54 days after the administration of the first dose. The meta-analysis on the RR of testing positive for any SARS-CoV-2 PCR included 18 entries and 14 studies. Menni et al. analyzed SARS-CoV-2 infection in individuals who received one or two doses of BNT162b2 (A) and ChAdOx1/AZD1222 (B) [43]. The results of the RE meta-analysis (IV) produced a significant RRR, 84%, for vaccinated individuals compared to unvaccinated individuals, with RR = 0.26. The test for heterogeneity was significant at 1% (Q = 1030.89, *p* < 0.0001) (Table 1, Figure 2c).

#### 3.1.2. RR of Symptomatic COVID-19 Infection following Vaccination

In order to assess the effectiveness of SARS-CoV-2 vaccination against symptomatic COVID-19 infection, we performed two meta-analyses, in partially and fully vaccinated individuals (Table 1). The meta-analysis on the full vaccination protocol included only studies estimating VE among individuals aged <69 years. The meta-analyses on VE against symptomatic SARS-CoV-2 infection pooled 9 studies and 17 entries in total (9 and 8 entries pertained to partial and full vaccination protocols, respectively). The IV method displayed significant results for both FE and RE (*p* < 0.05). According to the RE results (IV), partial vaccination achieved an RRR = 78% (95% CI = 31–93%) for symptomatic SARS-CoV-2 infection compared to unvaccinated, while fully vaccinated subjects exhibited a RRR = 94% (95% CI = 84–98%) (Figure 3a,b). FE estimates of RRR were lower for both partially and fully vaccinated individuals (RRR = 51% and RRR = 76%, respectively). Both FE and RE were statistically significant (*p* < 0.05).

#### 3.1.3. RR of Hospitalization Risk following Vaccination

Concerning hospitalization risk, an RE meta-analysis over two studies [22,31] produced an RR = 0.38 (95%CI = 0.2719; 0.5242) after a full BNT162b2 vaccination protocol. (Q = 2.04, *p* = 0.1529; I^2^ = 51.1% (0.0%; 87.6%)). Based on Bernal et al.’s data, at least one dose of the BNT162b2 vaccine ensured an RRR of death equal to 83%, compared to unvaccinated individuals [23].

Considering the substantial heterogeneity and the significance of the heterogeneity test, we could not be overly confident that the RR estimate would be robust in every context; therefore, we performed a sensitivity analysis to address the between-study heterogeneity (Appendix A).

#### 3.1.4. L’ Abbé Plot

The L’ Abbé plots in Figure 4 confirmed that, overall, infection events were in favor of the control group (unvaccinated). Concerning symptomatic SARS-CoV-2 infections, the L’Abbè plots in Figure 4d,e confirmed that the infection rates were greater in the unvaccinated groups than in the partially or fully vaccinated groups.

### 3.2. Subgroup Meta-Analysis

The GOSH analysis did not indicate a clear cluster that contributed to the pooled imbalance (Appendix A). In the subgroup meta-analysis, we grouped the studies by type of vaccine, mean age of sample (≥69 and <69 years), SARS-CoV-2 lineage, and bias assessment performed through Newcastle–Ottawa Scale (NOS). For simplicity, the NOS quality score was classified as satisfactory (NOS ≤ 6) and good (NOS > 6).

The FE subgroup meta-analysis partly explained the between-study heterogeneity because all the subgroups produced a significant Q test (*p* < 0.0001). However, considering the residual heterogeneity and the small number of studies in several subgroups, the results must be interpreted with caution.

The first subgroup analysis compared different vaccine technologies. Based on the FE subgroup estimates, the RRR after the first dose of BNT162b2 ranged from 46% to 49% for any positive SARS-COV-2 PCR and from 35% to 32% for symptomatic events. This discrepancy might be explained by the low number of studies included in the subgroup evaluating VE against symptomatic SARS-COV-2 infection (*n* = 6) with respect to the group evaluating the effectiveness of the BNT162b2 vaccine against any positive PCR (*n* = 16) (Table 2). In FE, the RR estimates were greater in higher-quality studies (NOS > 6), whether symptomatic or not (RR = 0.12 and RR = 0.07 respectively) compared to lower-quality studies (NOS ≤ 6, RR = 0.67 and RR = 0.48, respectively). Partial vaccination effectiveness appeared lower within studies that examined older populations (≥69 years) compared to younger, either on symptomatic PCR (RR = 0.40 and RR = 0.49, respectively) or any positive PCR (RR = 0.29 and RR = 0.48, respectively).

Regarding the FE subgroup meta-analysis in fully vaccinated individuals (Table 2), BNT162b2 produced an RRR = 83% against any positive PCR and an RRR = 68% against symptomatic SARS-CoV-2 infections. The RE and the mixed model produced an RRR of 94% against any positive PCR and an RRR of 84% against symptomatic COVID-19. Further, the FE subgroup meta-analysis relative to the bias assessment (NOS) showed a greater RR for lower-quality studies compared to higher-quality studies (RR = 0.01 in symptomatic PCR and an RR of 0.33 in any positive PCR meta-analyses, respectively). Full vaccination effectiveness against any positive PCR appeared slightly lower in studies that examined older populations (≥ 69 years) compared to younger populations (RR = 0.15 and RR = 0.12, respectively).

Concerning partial vaccination effectiveness against SARS-CoV-2 variants, the RRR of any positive PCR for B.1.1.7 was 60% in FE and 81% in the mixed model. The RRR increased to 85% in FE and 92% in the mixed model after full vaccination. Although only ‘Abu Raddad’ assessed the effectiveness of BNT162b2 against the B.1.351 lineage [25], the RRR was lower for the B.1.351 variant in both partially (RRR = 9%) and fully vaccinated subjects (RRR = 60%). Regarding symptomatic PCR, the RRR in fully vaccinated individuals remained greater for B.1.1.7 (RRR = 86%) compared to B.1.351 (RRR = 64%) (Table 2, a, b, e).

### 3.3. Publication Bias

The funnel plots are displayed in Appendix A. The contour-enhanced funnel plots do not exhibit publication bias. The Egger test indicated the presence of funnel plot asymmetry in the meta-analysis of symptomatic SARS-CoV-2 RR after the full vaccination protocol (intercept = −11.271, *p* = 0.026), although it may lack the statistical power to detect bias because the number of studies was small (*n* <10).

## 4. Discussion

Despite the considerable between-study heterogeneity, our findings provided evidence that any COVID-19 vaccine is highly effective outside the controlled conditions of clinical trials. Overall, SARS-CoV-2 vaccination is effective at reducing the number of new COVID-19 cases, either asymptomatic or symptomatic, with the greatest benefit achieved after completing the full vaccination protocol. Fewer real-world data were available for ChAdOx1/AZD1222 due to its later approval for deployment. However, vaccination with ChAdOx1/AZD1222 resulted in the rare onset of immune thrombotic thrombocytopenia; therefore, it underwent an additional careful monitoring process by European Union (EU) regulatory authorities [52].

According to our findings, partially vaccinated individuals were only a quarter as likely (RR = 0.26) to develop any SARS-CoV-2 infection as unvaccinated individuals, while their overall risk of symptomatic infection was slightly lower (RR = 0.22). The full vaccination schedule was 94% effective against asymptomatic and symptomatic positive PCR tests. Concerning hospitalization risk, a meta-analysis over two studies [22,31] yielded an RRR of 62% after the full BNT162b2 vaccination protocol. Concerning mortality risk, our findings corroborated Bernal et al.’s study [22], conducted over 7.5 million adults aged 70 years and older in the UK. At least one dose of BNT162b2 was approximately 83%, effective at preventing death compared to no vaccination, while there was insufficient follow-up to assess ChAdOx1/AZD1222’s impact on mortality because of its delayed rollout.

The subgroup analysis did not find any significant difference in effectiveness between the mRNA and the modified adenovirus vaccines, in part because the evidence, especially for ChAdOx1/AZD1222, was still very scarce. Up to 15 May, only two studies evaluated ChAdOx1/AZD1222 [22,23,41,42], and only one Ad26.COV2.S effectiveness [38]. As expected, COVID-19 VE appeared higher in adults aged <69 years. The evaluation of VE against the SARS-CoV-2 variants remained a key point that lacked robust results. In fact, only one study investigated the effectiveness of BNT162b2 against two variants of concern [25]. The RR of infection following the full vaccination protocol appeared larger for the B.1.351 (South Africa) variant compared to B.1.1.7 (UK). Although the evidence suggested a lower VE against SARS-COV-2 variants of concern, our findings were not sufficient to assert that full COVID-19 vaccination protocol is 60% effective against B.1.351 and 94% effective against B.1.1.7.

Although imprecise, the real-world data confirmed the experimental evidence and suggested that the vaccine offers mild protection against emerging SARS-CoV-2 variants as well as within the elderly population. By including non-randomized studies, our research aimed to address questions not answered by clinical trials, such as VE within different population subgroups. Moreover, the effect of waning immunity on effectiveness needs long-term investigation, which appears more compatible with a non-RCT study design.

The present study features several limitations. The subgroup analysis and the between-study heterogeneity analysis were not able to reduce the overall heterogeneity. The sparsity of data at patient level did not allow further investigation of unobserved sources through a metaregression. Undeniably, the heterogeneity may have stemmed from the observational design of the included studies and, unless through a randomized process, it is unlikely to be reduced in any circumstances.

Five studies did not test their vaccinated cohorts systematically during follow-up. The absence of active laboratory surveillance of vaccinated individuals might have resulted in an underestimation of asymptomatic cases. However, we did not consider a different rate of testing as a bias of concern in terms of RR estimate because both vaccinated and unvaccinated individuals underwent the same case investigation and contact tracing protocol. In addition, when systematic testing was not performed, asymptomatic testing was available to different extents: workplace exposures (HCWs), out-of-state travelers or per-request [28].

Ultimately, asymptomatic cases not confirmed by PCR test, as well as false negatives, might represent a source of bias in both vaccinated and unvaccinated groups [28,34]. Finally, public health mitigation measures might contribute to the underestimation of SARS-CoV-2 vaccine effectiveness. Unfortunately, the extent to which primary prevention restrictions affected VE was beyond the scope of this study.

## 5. Conclusions

By way of conclusion, we can state that a significant reduction in the RR of asymptomatic infection within partially vaccinated individuals was not corroborated by sufficient statistical robustness for the results to be generalizable. However, full vaccination effectiveness against symptomatic and asymptomatic SARS-CoV-2 infection risk confirmed the RCT results, leading to the same RR estimates but larger CI. In order to investigate additional sources of heterogeneity that might affect the validity of the meta-analysis results, further research on real-world SARS-CoV-2 vaccine effectiveness is encouraged. Ultimately, our findings support the maximization of full vaccination coverage. Additional evidence about the impact of SARS-CoV-2 variants on vaccine effectiveness is vital in order to monitor mutations associated with vaccine escape.

## Figures and Tables

**Figure 1 vaccines-10-00157-f001:**
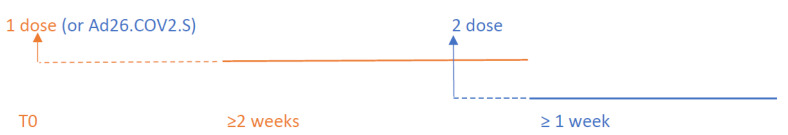
Experimental group follow-up: the dashed lines following the first and the second dose administration represent the induction time. The solid lines display the follow-up period considered in the analysis. Cases occurred within 14 days after the first dose uptake were not included. Cases occurred within one week from the second dose administration were attributed to the first dose effect. The length of the induction time for the Ad26.COV2.S vaccine was set at 14 days after the single dose administration (details in the text).

**Figure 2 vaccines-10-00157-f002:**
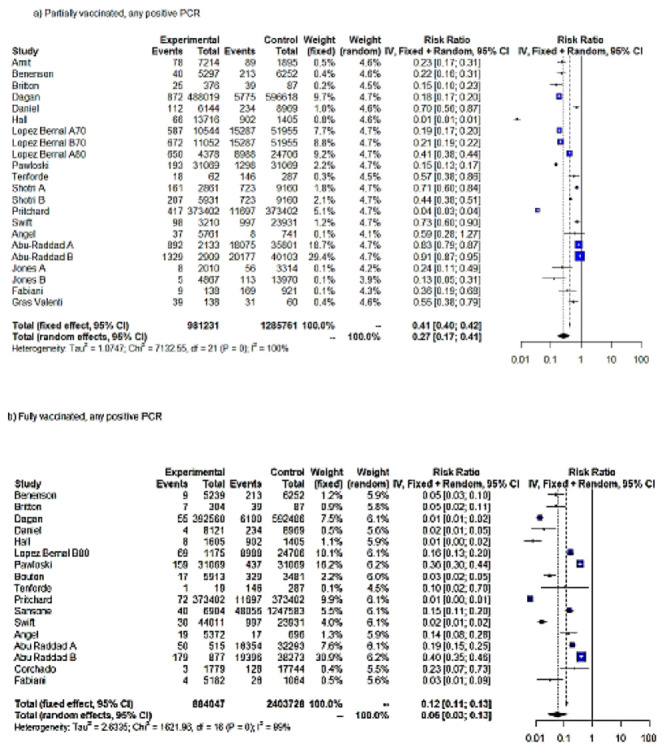
Forest plot, partially vaccinated. Any positive SARS-CoV-2 PCR RR (95%-CI) ≥ 14 days from first dose uptake (**a**). Forest plot, fully vaccinated. Any positive PCR RR (95%-CI) ≥ 7 days from full vaccination (**b**). Forest plot, any positive PCR RR (95%-CI), ≥14 days from vaccination with at least one dose. IV method (**c**).

**Figure 3 vaccines-10-00157-f003:**
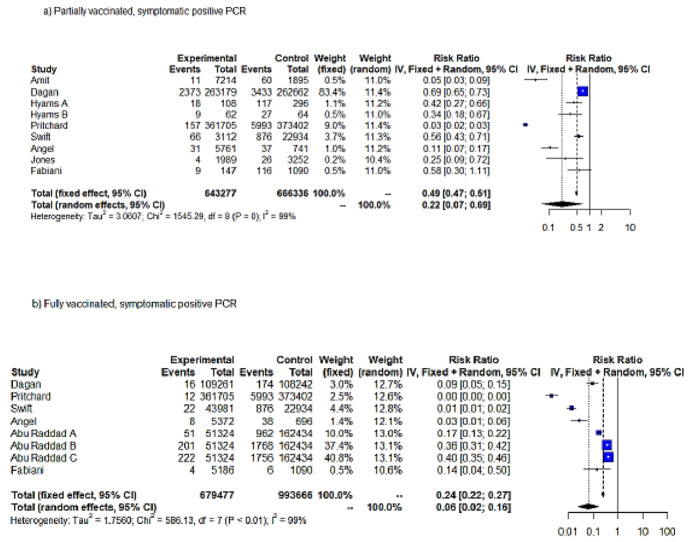
Forest plot, symptomatic positive SARS-CoV-2 PCR RR (95%-CI) ≥ 14 days from first dose uptake (**a**). Forest plot, fully vaccinated. Symptomatic positive PCR RR (95%-CI) ≥ 7 days from full vaccination (**b**). IV method.

**Figure 4 vaccines-10-00157-f004:**
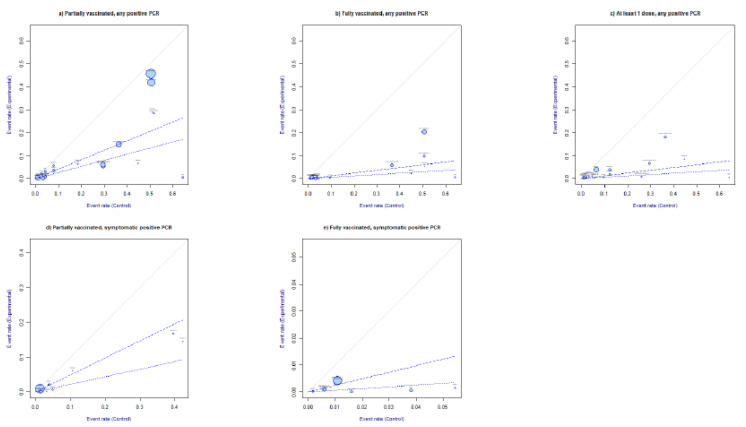
L’Abbè plots. Partially vaccinated, any positive SARS-CoV-2 PCR RR ≥ 14 days from first SARS-CoV-2 dose administration (**a**). Fully vaccinated, any positive PCR RR, ≥7 days from full vaccination (**b**). At least one dose, any positive PCR RR ≥ 14 days from vaccination with first dose (**c**). Partially vaccinated, symptomatic positive PCR RR ≥ 14 days from first dose administration (**d**). Fully vaccinated, symptomatic positive SARS-CoV-2 PCR RR ≥ 7 days from full vaccination (**e**). The sizes of the plotted circles are proportional to the precision of the studies. The dashed lines mark the overall estimate of the log risk for RE (light) and FE (bold). The further the circle from the line of no effect, the greater the difference of event rates between intervention and control arms.

**Table 1 vaccines-10-00157-t001:** Meta-analyses: relative risk of SARS-CoV-2 on positive PCR test after vaccination. Fixed (FE) and Random Effect (RE), inverse variance method (IV) and Mantel–Haenszel (MH) method. RR of any positive PCR and symptomatic positive PCR after SARS-CoV-2 vaccination. Any SARS-CoV-2 vaccination protocol.

Method	PCR Test	SARS-CoV-2 Vaccination Protocol	N	Fixed Effect Model		Random Effect Model		τ^2^	I^2^
				RR [95%-CI]	*p*-Value	RR [95%-CI]	*p*-Value		
Inverse Variance (IV)	Any positive PCR	Partially vaccinated	22	0.4115 [0.4025; 0.4207]	0	0.2657 [0.1710; 0.4127]	<0.0001 *	1.0747 [0.6223; 3.0350]	99.70%
		Fully vaccinated	17	0.1204 [0.1120; 0.1295]	0	0.0586 [0.0266; 0.1292]	<0.0001 *	1.6228 [1.0654; 2.4896]	99.00%
		At least one dose	18	0.3813 [0.3752; 0.3875]	0	0.1617 [0.1130; 0.2313]	<0.0001 *	0.5862 [0.4444; 2.6714]	99.70%
	Symptomatic positive PCR test	Partially vaccinated	9	0.4885 [0.4658; 0.5122]	<0.0001	0.2181 [0.0685; 0.6944]	0.01 *	3.0607 [0.7297; 10.0860]	99.50%
		Fully vaccinated	8	0.2439 [0.2231; 0.2666]	<0.0001	0.0629 [0.0245; 0.1613]	<0.0001 *	1.7560 [1.1035; 14.6961]	98.80%
Mantel–Haenszel (MH)	Any positive PCR	Partially vaccinated	22	0.2490 [0.2436; 0.2546]		0.2656 [0.1617; 0.4363]		1.374	
		Fully vaccinated	17	0.0447 [0.0419; 0.0477]		0.0586 [0.0228; 0.1505]		3.7986	
		At least one dose	18	0.3369 [0.3315; 0.3424]		0.1616 [0.1122; 0.2328]		0.6083	
	Symptomatic positive PCR test	Partially vaccinated	9	0.2758 [0.2644; 0.2878]	0	0.2182 [0.0567; 0.8388]	0.0267 *	4.1669	99.60%
		Fully vaccinated	8	0.0811 [0.0752; 0.0874]	0	0.0626 [0.0167; 0.2344]	<0.0001 *	3.5394	99.40%

* significant results (*p* < 0.05).

**Table 2 vaccines-10-00157-t002:** Subgroup meta-analysis, any SARS-CoV-2 protocol. Fixed-effect model.

Subgroups		N	Results for Subgroups			Between-Group Heterogeneity		
			RR (95%-CI)	I^2^	tau2	Q	df(Q)	I-Value
**(a) Partially Vaccinated, Any Positive PCR—Fixed Effect**								
Vaccine	BNT162b2	16	0.5250 (0.5123;0.5381)	99.60%	0.7549	3108.43	3	0
	ChAdOx1/AZD1222	2	0.2277 (0.2122;0.2443)	98.90%	0.3565			
	BNT162b2/mRNA-1273	3	0.2778 (0.2474;0.3120)	98.80%	1.0692			
	BNT162b2/ChAdOx1/AZD1222	1	0.0357 (0.0323;0.0393)	-				
Quality	NOS ≤ 6	15	0.4816 (0.4704;0.4931)	99.60%	0.6113	1502.28	1	0
	NOS > 6	7	0.1219 (0.1142;0.1302)	99.70%	3.3748			
Age	<69 years	15	0.4828 (0.4702;0.4958)	99.80%	1.5669	463.24		<0.0001
	≥69 years	7	0.2845 (0.2733;0.2962)	98.60%	0.2424			
Lineage	B.1.1.7	8	0.3903 (0.3777;0.4033)	99.70%	0.9447	3031.61	3	0
	B.1.1.7/non-B.1.1.7	5	0.1492 (0.1418;0.1569)	99.70%	1.4303			
	Not specified	8	0.3254 (0.2971;0.3565)	97.10%	0.6506			
	B.1.351	1	0.9080 (0.8717;0.9458)	-				
**(b) Fully Vaccinated, Any Positive PCR—Fixed Effect**								
Vaccine	BNT162b2	11	0.1680 (0.1537; 0.1836)	98.40%	1.7775	702.84	3	<0.0001
	BNT162b2/mRNA-1273	4	0.1624 (0.1394; 0.1893)	98.90%	3.7392			
	BNT162b2/ChAdOx1/AZD1222	1	0.0062 (0.0049; 0.0078)	-	-			
	Ad26.COV2.S	1	0.2338 (0.0745; 0.7337)	-	-			
Quality	NOS ≤ 6	11	0.1947 (0.1797; 0.2108)	98.50%	1.3935	827.35	1	<0.0001
	NOS >6	6	0.0112 (0.0094; 0.0134)	95.90%	1.6474			
Age	<69 years	13	0.1171 (0.1084; 0.1266)	99.30%	3.0726	4.21	1	0.0403
	≥69 years	4	0.1487 (0.1200; 0.1843)	65.00%	0.2931			
Lineage	B.1.1.7	4	0.1448 (0.1231; 0.1703])	95.90%	0.817	1000.54	3	<0.0001
	B.1.1.7/non-B.1.1.7	5	0.0190 (0.0165; 0.0219)	98.60%	1.9969			
	Not specified	7	0.1692 (0.1452; 0.1971)	97.70%	3.4732			
	B.1.351	1	0.4027 (0.3533; 0.4592)	-	-			
**(c) At Least One Dose, Any Positive PCR—Fixed Effect**								
Vaccine	BNT162b2	12	0.2575 (0.2511; 0.2640)	99.60%	0.6218	2752.22	2	0
	BNT162b2/mRNA-1273	4	0.5641 (0.5517; 0.5767)	98.50%	0.4039			
	ChAdOx1/AZD1222	2	0.1462 (0.1354; 0.1578)	83.10%	0.2114			
Quality	NOS ≤6	16	0.2550 (0.2491; 0.2610)	99.40%	0.432	2182.95	1	0
	NOS >6	2	0.5525 (0.5402; 0.5650)	99.90%	8.9219			
Age	<69 years	12	0.2196 (0.2133; 0.2260)	99.30%	0.5076	2070.75	1	0
	≥ 69 years	6	0.4927 (0.4832; 0.5025)	99.80%	0.5585			
Lineage	B.1.1.7	4	0.3357 (0.3223; 0.3497)	99.80%	0.8863	1015.58	2	<0.0001
	B.1.1.7/non-B.1.1.7	5	0.1463 (0.1372; 0.1560)	95.00%	0.3366			
	Not specified	9	0.4231 (0.4154; 0.4310)	99.80%	0.5975			
**(d) Partially Vaccinated, Symptomatic Positive PCR—Fixed Effect**								
Vaccine	BNT162b2	6	0.6572 (0.6245; 0.6915])	99.60%	1.1044	1414.28	3	<0.0001
	ChAdOx1/AZD1222	1	0.3441 (0.1763; 0.6715)	-				
	BNT162b2/mRNA-1273	1	0.5552 (0.4336; 0.7111)	-				
	BNT162b2/ChAdOx1/AZD1222	1	0.0270 (0.0231; 0.0317)	-				
Quality	NOS ≤6	6	0.6688 (0.6354; 0.7039)	99.60%	0.7447	1057.05	1	<0.0001
	NOS >6	3	0.0676 (0.0595; 0.0769)	99.50%	3.5694			
Age	<69 years	7	0.4902 (0.4673; 0.5143)	99.60%	3.384	1.25	1	0.2636
	≥69 years	2	0.3962 (0.2736; 0.5737)	0.00%	-			
Lineage	B.1.1.7	3	0.2410 (0.1801; 0.3224)	89.10%	0.5695	26.49	3	<0.0001
	B.1.1.7/non-B.1.1.7	2	0.5032 (0.4789; 0.5287)	99.90%	5.242			
	Not specified	4	0.4116 (0.3326; 0.5094)	94.10%	1.3278			
**(e) Fully Vaccinated, Symptomatic Positive PCR—Fixed Model**								
Vaccine	BNT162b2	6	0.3181 (0.2901; 0.3489)	94.90%	0.3245	488.27	2	<0.0001
	BNT162b2/ChAdOx1/AZD1222	1	0.0021 (0.0012; 0.0036)	-	-			
	BNT162b2/mRNA-1273	1	0.0131 (0.0086; 0.0200)	-	-			
Quality	NOS ≤ 6	5	0.3301 (0.3008; 0.3622)	93.00%	0.1911	492.15	1	<0.0001
	NOS > 6	3	0.0085 (0.0063; 0.0116)	94.60%	1.4624			
Lineage	B.1.1.7	2	0.1347 (0.1034; 0.1753)	94.80%	1.565	222.39	3	<0.0001
	B.1.1.7/non-B.1.1.7	3	0.2744 (0.2408; 0.3127)	99.40%	6.8062			
	Not specified	2	0.0166 (0.0111; 0.0248)	91.80%	2.5779			
	B.1.351	1	0.3598 (0.3111; 0.4162)	-	-			

## Data Availability

Data supporting the reported results are available on request to the Authors.

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
