# Peer review of "Effectiveness of Vaccination against SARS-CoV-2 Infection in the Pre-Delta Era: A Systematic Review and Meta-Analysis"

_vaccines, 2022, doi:10.3390/vaccines10020157_

Round 1

Reviewer 1 Report

The post-licensure effectiveness of vaccination against Covid-19 infection and disease has been studied in a large number of individual reports. This article has pooled 27 such studies for meta-analyses. It is difficult to judge what the value of such a meta-analysis is.

The meta-analysis focuses on early stage of vaccine introduction, from March to May 2021, and effectiveness against the then prevalent SARS-CoV-2 virus variants (UK, B. 1.1.7). This information is nearly obsolete. The short term effectiveness of one or two doses is well documented in individual studies. Now the key issue is long term effectiveness, i.e. duration of protection, which is not addressed in this study. Moreover, of particular interest is the vaccine effectiveness in vulnerable age groups. The present study makes an attempt to divide some studies by age less than 69 years of age and over, which is not good enough. Furthermore, the real life issue is vaccine effectiveness against delta variant, which is recognized by the authors but not addressed in the study.

The writing is dry official style and difficult to read. So it is not unlikely that any ordinary doctors would want to read this paper or benefit from it. Therefore, it remains a bureucratic exercise with little practical application.

Author Response

THe post-licensure effectiveness of vaccination against Covid-19 infection and disease has been studied in a large number of individual reports. This article has pooled 27 such studies for meta-analyses. It is difficult to judge what the value of such a meta-analysis is.

RE: Any meta-analysis has a value (and heterogeneity attached). This piece of work was concluded in July 2021, while the earliest mass vaccination campaign was reaching the largest part of worldwide population. The first submission took 4 months to conclude the peer reviewing process, meanwhile the most widespread Covid-19 variant of concern has changed. In general, the overall estimates of the meta-analysis are fundamental, as baseline to address future research in vaccine effectiveness and to adjust the methodology. Furthermore, policy makers require the support of scientific evidence ( including the uncertainty and the marginal probability of errors) before making decisions in public health realm. Overall, the meta-analysis provides baseline estimates and allows to understand how the Sars-Cov2 (and the vaccines effectiveness, respectively) is evolving. Research validation takes time, especially meta-analysis, therefore, the utility of this job is mostly focused on addressing the future methodology on vaccine effectiveness and to support public health strategies (as well as pharma strategies) on balancing mass vaccination campaigns with mitigation measures’ spells .

The meta-analysis focuses on early stage of vaccine introduction, from March to May 2021, and effectiveness against the then prevalent SARS-CoV-2 virus variants (UK, B. 1.1.7). This information is nearly obsolete. The short term effectiveness of one or two doses is well documented in individual studies. Now the key issue is long term effectiveness, i.e. duration of protection, which is not addressed in this study.

AM: Please see the comment above. Individual studies, even large cohort studies, do not have the power of meta-analysis. Moreover, the sentence ‘The short term effectiveness of one or two doses is well documented in individual studies. Now the key issue is long term effectiveness, i.e. duration of protection, which is not addressed in this study’ contains two mistakes at least, hence, it stem from wrong assumptions:
1.Long term effectiveness, as well as the epidemic R, cannot be tested if you are not able to separate two parameters at least: the virus virulence and the waning immunity, which is time dependent. In other words, is the vaccine less effective because of the new variant or because of the waning immunity? One possibility to test that, for instance, would be to compare vaccine effectiveness against the same Sars-Cov-2 variant over time, whereas that’s impossible because the virus mutates very quickly. In alternative, one can booster the vaccination and measure effectiveness against a different variant. You still need the vaccine effectiveness against the This might reduce the uncertainty due to time dependency.
2.What does ‘Individual studies’ mean? It’s meant quasi-experimental? Cohort? RCT? What kind of study design is the reviewer referring to? RCTs do not have the same external validity as meta-analysis does. Yet, as any researcher is supposed to know, meta-analysis is the top of scientific evidence and has more statistical power than large cohort studies. An health policy decision cannot be based on one or more studies, they are not comparable, unless a statistical synthesis between them, which is meta-analysis.

Moreover, of particular interest is the vaccine effectiveness in vulnerable age groups. The present study makes an attempt to divide some studies by age less than 69 years of age and over, which is not good enough. Furthermore, the real life issue is vaccine effectiveness against delta variant, which is recognized by the authors but not addressed in the study.

RE:

This comment is not helpful. Please explain what do you mean by ‘not good enough’. The mass vaccination campaign identified age group priority(>80; >70; healthcare workers…), and data availability was very limited. Some studies tested vaccine effectiveness only among elderly (≥70), some others within the entire population aged ≥18.  In order to reduce heterogeneity, we split the  subgroups between studies enrolling older and younger than 69 years. The aim of subgroup analysis is to reduce/adress uncertainty, not to find causality! You cannot use patient level characteristics for subgroup analysis and it doesn’t mean any causality! In order to assess vaccine effectiveness in vulnerable groups, you’d need a meta-regression, for example, which was not the scope of this study. Even in this scenario, data at patient level were too sparse so meta-regression was not feasible.

The writing is dry official style and difficult to read. So it is not unlikely that any ordinary doctors would want to read this paper or benefit from it. Therefore, it remains a bureaucratic exercise with little practical application.

RE: The authors are actually doctors. However, we amended the abstract, making it ‘ordinary doctor friendly’. However, the main target of meta-analysis in general, and this meta-analysis in particular, are decision-makers (in this specific case pertaining to the public health realm). Nevertheless, doctors are supposed to understand results and conclusions of a meta-analysis, or somehow, such a basic statistics. Ultimately, we would like to disregard the last part of the comment because it appears as a personal opinion rather than an objective comment. The term ‘bureaucracy’ should be used in a different context, this is known as evidence based method.

Reviewer 2 Report

The study presents a large statistical analysis of the efficacy of the various  vaccines on the transmission. It concludes that all vaccines are very effective. This is a very important study. 

General comments

A colossal work has been done. Sadly, the way it is presented  makes it impenetrable for the general readership. For example,  I am a virologist and modeler of virus evolution and immunology,  not a biostatistician. I am really interested in this topic, but cannot read the MS. What  about the other readers?

1. Firstly, I did not find any figures in the main text, only their placeholders.
Figures in the "Annex" are not the figures in the main text.

2. The frequent repeat of long technical names of vaccines puts the reader asleep in a minute. Unless this is the goal of the authors, and they prepared an ASMR piece, why not to use commercial names, defining the technical names  them once in the beginning? 

3. This sentence is  incorrect. Vaccines do not block transmission, as stated by the authors one sentence earlier:

The current licensed vaccines may be transmission   blocking, especially after full vaccination protocol

4. Abstract is full of technical biostatistical terms and not accessible to the broad virological and immunological readership:
A frequentist random effects 12 meta-analysis was carried out.

subgroup meta-analysis;

quantitative synthesis. 

I would recommend complete rewriting, and definitely with the figures attached to the main text, preferably in the middle of the text.  I would even consult someone who does scientific writing.

Author Response

REV 2

The study presents a large statistical analysis of the efficacy of the various  vaccines on the transmission. It concludes that all vaccines are very effective. This is a very important study.

General comments

A colossal work has been done. Sadly, the way it is presented  makes it impenetrable for the general readership. For example,  I am a virologist and modeler of virus evolution and immunology,  not a biostatistician. I am really interested in this topic, but cannot read the MS. What  about the other readers?

RE: Such an important study requires advanced statistics. In particular, the expected heterogeneity needs to be carefully investigated. That’s mandatory, by meta-analysis guidelines. If  you are a modeler, as you claimed,  you should know that each epidemic curve requires a baseline RR to draw out the probability of infection (just an example..). Ultimately, we may attempt to make the abstract more ‘ordinary reader  friendly’ (see above), but it’s more likely that an epidemiologist, a virologist or an immunologist will be interested in this journal and this paper. Honestly, it’ s the first time we are receiving such a comment. The methodology and results of meta-analysis should be of public domain nowadays.

  1. Firstly, I did not find any figures in the main text, only their placeholders. Figures in the "Annex" are not the figures in the main text.

RE: That’s not on us. It depends on the submission guidelines.

  1. The frequent repeat of long technical names of vaccines puts the reader asleep in a minute. Unless this is the goal of the authors, and they prepared an ASMR piece, why not to use commercial names, defining the technical names them once in the beginning?

RE: At the time of meta-analysis/systematic review the commercial names of the Covid-19 vaccines were not available yet. Many of the included studies used technical names.

  1. This sentence is incorrect. Vaccines do not block transmission, as stated by the authors one sentence earlier: ‘The current licensed vaccines may be transmission blocking, especially after full vaccination protocol’

RE: The sentence is not incorrect but we amended that statement in the abstract, because it was misleading.(l.26-31) The results suggest that, given the assumptions made in the methodology, including the time horizon, the full vaccination schedule might be transmission blocking! Unfortunately, the high heterogeneity was not reduced by the extended analysis. (l.307) Therefore, the final result is not generalizable unless further research and data availability. The reason underlying the transmission blocking effect are not known, we can just make hypothesis on that. Obviously, there must be some unobserved effect that could be captured by patient level characteristics, comorbidities, or waning immunity(l.416-427). However, this was not possible to be modelled, because of sparse data. Moreover, that is beyond the research scope. In this light, we encouraged further research in experimental field that could suggest a possible explanation. That would help also to reduce uncertainty surrounding our findings.(l.417-419)

  1. Abstract is full of technical biostatistical terms and not accessible to the broad virological and immunological readership:

-A frequentist random effects 12 meta-analysis was carried out.

-subgroup meta-analysis;

-quantitative synthesis.’

I would recommend complete rewriting, and definitely with the figures attached to the main text, preferably in the middle of the text.  I would even consult someone who does scientific writing.

RE: 

Those terms are very general terms in medicine as well as in epidemiology. In particular, they are commonly used in meta-analysis, as well as in the scientific literature. They are not interchangeable with synonyms. However, we amended part of the abstract where those terms were present (l. 12-17; 26-31) and we reformulated the sentences in the discussion (392-395 and 406-408) to improve readability.

Further changes:

-l.304-305: we specified that papers were classified in two categories according to NOS quality score.

Round 2

Reviewer 1 Report

-

Author Response

Minor english language revision have been integrated

Author Response

english language style and check have been performed
